# The E3 Ubiquitin Protein Ligase LINCR Amplifies the TLR-Mediated Signals through Direct Degradation of MKP1

**DOI:** 10.3390/cells13080687

**Published:** 2024-04-15

**Authors:** Takumi Yokosawa, Sayoko Miyagawa, Wakana Suzuki, Yuki Nada, Yusuke Hirata, Takuya Noguchi, Atsushi Matsuzawa

**Affiliations:** Laboratory of Health Chemistry, Graduate School of Pharmaceutical Sciences, Tohoku University, Sendai 980-8578, Japan

**Keywords:** Toll-like receptors (TLRs), MAP kinases, MAPK phosphatase-1 (MKP1), the E3 ubiquitin ligase LINCR, ubiquitination

## Abstract

Toll-like receptors (TLRs) induce innate immune responses through activation of intracellular signaling pathways, such as MAP kinase and NF-κB signaling pathways, and play an important role in host defense against bacterial or viral infections. Meanwhile, excessive activation of TLR signaling leads to a variety of inflammatory disorders, including autoimmune diseases. TLR signaling is therefore strictly controlled to balance optimal immune response and inflammation. However, its balancing mechanisms are not fully understood. In this study, we identified the E3 ubiquitin ligase LINCR/ NEURL3 as a critical regulator of TLR signaling. In LINCR-deficient cells, the sustained activation of JNK and p38 MAPKs induced by the agonists for TLR3, TLR4, and TLR5, was clearly attenuated. Consistent with these observations, TLR-induced production of a series of inflammatory cytokines was significantly attenuated, suggesting that LINCR positively regulates innate immune responses by promoting the activation of JNK and p38. Interestingly, our further mechanistic study identified MAPK phosphatase-1 (MKP1), a negative regulator of MAP kinases, as a ubiquitination target of LINCR. Thus, our results demonstrate that TLRs fine-tune the activation of MAP kinase pathways by balancing LINCR (the positive regulator) and MKP1 (the negative regulator), which may contribute to the induction of optimal immune responses.

## 1. Introduction

Innate immune responses are critical for host defense against pathogens to eliminate them by inducing an inflammatory response. However, excessive induction of immune responses frequently leads to overproduction of pro-inflammatory cytokines (a cytokine storm), which can cause severe symptoms such as thrombus formation and multiorgan failure [1]. Pattern recognition receptors (PRRs) that are essential for initiating innate immune responses recognize unique and common structures of pathogens as pathogen-associated molecular patterns (PAMPs) and induce inflammatory responses [2,3,4]. Toll-like receptor 4 (TLR4), one of the PRRs, recognizes lipopolysaccharide (LPS)—a component of the cell walls of Gram-negative bacteria—and promotes the production of inflammatory cytokines mediated by phosphorylation-signaling pathways [4,5,6,7]. Therefore, elucidation of the detailed mechanisms of TLR4 signaling is expected to lead to development of new therapeutic strategies for severe infectious and autoimmune diseases [6,7]. TLR4 signaling is strictly controlled by post-translational modifications of the protein such as phosphorylation and ubiquitination [4,5,6,7]. When TLR4 binds to the ligand, it forms a signaling complex that includes the E3 ubiquitin ligase TNF receptor-associated factor 6 (TRAF6), which triggers K63-auto-polyubiquitination of TRAF6 [4,5,8]. Polyubiquitinated TRAF6 recruits TGF-β-activated kinase1 (TAK1)-binding protein 2 (TAB2) and TAB3 to form the TAK1–TAB2/3 complex, which triggers TAK1 activation by enhancing the autophosphorylation of TAK1 [9]. Subsequently, TAK1 leads to the activation of mitogen-activated protein kinase (MAPK) and nuclear factor-kappa B (NF-κB), resulting in the production of pro-inflammatory cytokines [4,5,8,10]. Meanwhile, suppressive mechanisms to dampen immune responses mediated by TLR4 signaling have been identified [11]. For instance, A20 and cylindromatosis (CYLD) induced by TLR4 signaling suppress downstream signaling by cleaving the K63-linked polyubiquitin chain of TRAF6 as deubiquitinating enzymes [11,12,13].

Mitogen-activated phosphatase-1 (MKP1), also known as dual specificity phosphatase-1 (DUSP1), is a suppressor of TLR signaling [11]. In general, MKP1 dephosphorylates MAP (mitogen-activated protein) kinases, such as p38 MAP kinase, c-Jun-N-terminal kinase (JNK), and extracellular signal-regulated kinase (ERK), and negatively regulates the production of pro-inflammatory cytokines [14,15]. Under steady-state conditions, the expression of MKP1 is kept at low levels, while mRNA is induced upon ERK activation [16]. In addition, ERK also regulates the expression of MKP1 at the protein level through ERK-dependent phosphorylation of MKP1 that inhibits the proteasomal degradation of MKP1 [17,18,19,20,21]. Moreover, S-phase kinase-associated protein (Skp2) and atrogin-1 have been identified as E3 ligases that degrade MKP1 upon serum-induced activation of ERK and during myocardial ischemia and reperfusion, respectively [20,21]. However, the E3 ligase that degrades MKP1 in innate immune responses is not identified.

Lung-inducible neuralized-related C3HC4 RING domain protein (LINCR), also known as neuralized E3 ubiquitin protein ligase 3 (NEURL3), has been originally identified as an E3 ligase induced by exposure of alveolar epithelial cells in mice to LPS [22,23,24]. Further studies have suggested that LINCR is involved in lung development [25] and spermatogenesis [26], although its biological functions remain unknown. In addition, recent evidence has demonstrated that LINCR induced by hepatitis C virus (HCV) infection suppresses HCV production [27] and that LINCR induced by Sendai virus infection triggers K63-linked polyubiquitination on IRF7, which in turn augments host antiviral immune response [28]. However, the functions of LINCR in TLR4 signaling remain unknown, even though LINCR has been identified as the E3 ubiquitin ligase induced by TLR4 activation.

In this study, we found a novel role of LINCR in TLR signaling. LINCR promotes the activation of MAP kinase pathways through the K48-linked polyubiquitination and subsequent proteasomal degradation of MKP1, leading to the enhanced production of pro-inflammatory cytokines. Therefore, LINCR appears to promote immune response against bacterial infection.

## 2. Materials and Methods

### 2.1. Cell Culture and Transfection

HEK293, Phoenix-AMPHO (ATCC) cells, and RAW264.7 cells were cultured in Dulbecco’s Modified Eagle’s medium and RPMI 1640 medium, respectively, including 1% penicillin–streptomycin solution and 10% heat-inactivated fetal bovine serum in 5% CO_2_ at 37 °C. Plasmid transfection was conducted by using Polyethylenimine “Max” (PEI-MAX)( Polysciences, Warrington, PA, USA), according to the manufacturer’s instructions.

### 2.2. Plasmids

Samples of cDNAs encoding human LINCR and MKP1 were obtained by conducting PCR and were inserted into pcDNA3 with FLAG or 6Myc tag plasmids. FLAG-LINCR-2CS (^202^Cys and ^205^Cys were changed to Ser) and FLAG-LINCR-ΔRING (lacking 202-241) were obtained by performing PCR. FLAG-MKP1 was subcloned into pGEX6P-1.

### 2.3. Antibodies and Reagents

All reagents were purchased from commercial sources; LPS (InvivoGen, San Diego, CA, USA), SB203580 (Wako, Osaka, Japan), SP600125 (Wako), U0126 (Wako), 5z-7 (Santa Cruz, Dallas, TX, USA), ML120B (Sigma, St. Louis, MO, USA), TPL2 kinase inhibitor (Cayman Chemical, Ann Arbor, MI, USA), MG132 (Santa Cruz), poly(I:C) (InvivoGen), and Flagellin (InvivoGen). Antibodies against the following proteins were used in this study: p38, phospho-p38, JNK, phospho-JNK, ERK, phospho-ERK, p65, MKP1, MKP5, K48-linked polyubiquitin (Cell Signaling, Danvers, MA, USA), Myc-562 (MBL, Tokyo, Japan), FLAG-1E6 (Sigma), FLAG-M2 (Wako), HA (Roche, Basel, Switzerland), and PARP1, Myc-9E10, and β-actin (Santa Cruz). Mouse IgG, rat IgG, and rabbit IgG (Cell Signaling) were also used in this study. Anti-LINCR was produced by Cosmobio (Tokyo, Japan) using a part of the mouse-LINCR sequence (CDDQRSTARRRSTFHDGIV) as the antigen.

### 2.4. Generation of KO Cells

LINCR, MKP1, and MKP5 KO cells were established by using the CRISPR/Cas9 system as previously described [29]. Guide RNAs (gRNAs)were designed to target the region in the exon 2 of LINCR gene (5′- GCGAGGCCCTTAGTTTCCACGGG-3′), the exon 2 of MKP1 gene (5′-AGTACCCCTCTCTACGATCAGG-3′), and the exon 2 of MKP5 gene (5′-CGATAAGATCAGCCGGCGAAGG-3′) by using CRISPRdirect. Oligonucleotide encoding gRNA was cloned in lentiCRISPRv2 plasmid (Addgene, Watertown, MA, USA), and knockout cells were generated as previously described [29]. To check the mutations of the genes in cloned cells, a genomic sequence was conducted by using PCR-direct sequencing. Extracted DNA from each clone was used as a template along with the following primers: 5′-TACCATCGTGGAACGTCGAG-3′ and 5′-CCTGGGGCTTGTGAGATGAG-3′ for LINCR; 5′-GAAGCGTTTTCGGCTTCCTG-3′ and 5′-GAGCCAACAAGCTCTTCCGT-3′ for MKP1; and 5′-CCCCAATGACCTGGCAAAGA-3′ and 5′-TTCATGGTGCCTTGGGGTTA-3′ for MKP5.

### 2.5. Stable Cell Lines

Stable cell lines that express TLR4 or FLAG-LINCR were generated by retroviral transduction as follows [30]. Phoenix-AMPHO (the packaging cell line) was transfected with pMXs-IH inserted with either TLR4, or FLAG-LINCR WT or ΔRING. After 48 h, the growth medium that contained retrovirus was collected. HEK293 and RAW264.7 cells were incubated with the (virus-containing) medium with 10 µg/mL polybrene for 48 h, and uninfected cells were eliminated through hygromycin selection.

### 2.6. Immunoblotting

Cells were lysed with DISC lysis buffer TX (1% Triton X-100, 150 mM NaCl, 20 mM Tris-HCl (pH 7.4), 1% protease inhibitor cocktails (Nacalai Tesque, Kyoto, Japan), and 10% Glycerol). After centrifugation, the cell extracts were resolved by SDS-PAGE, and the blots were developed with Immobilon ECL Ultra Western HRP Substrate (Merck, Darmstadt, Germany).

### 2.7. Nuclear Extraction

Cells seeded on 6-well plates at a density of 5 × 10^5^/well were lysed in ice-cold lysis buffer containing 10 mM KCl, 0.1 mM EDTA, 0.1 mM EGTA, 10 mM Hepes (pH 7.5), 1 mM DTT, and 1% protease inhibitor cocktails (Nacalai) for 15 min, after indicated stimulation or treatment. Cell lysates with an added 1% NP-40 were centrifuged at 2500 rpm for 3 min at 4 °C. After the supernatants that contain cytoplasmic fractions were removed, the pellets were suspended in ice-cold lysis buffer containing 1 mM EGTA, 400 mM NaCl, 20 mM Hepes (pH 7.5), 1 mM DTT, and 1% protease inhibitor cocktails for 15 min and were vortexed every 5 min. Cell lysates were then centrifuged at 15,000 rpm for 15 min at 4 °C, and then the supernatants were collected as nuclear fraction.

### 2.8. Immunoprecipitation

Immunoprecipitation was carried out as previously described [31]. The cells were lysed in the lysis buffer. After centrifugation, their supernatants were immunoprecipitated with anti-FLAG affinity M2 gel (Sigma) or anti c-Myc antibody beads (Wako). The immunoprecipitates were subsequently washed with lysis buffer and subjected to immunoblot analysis.

### 2.9. Recombinant Protein Purification

HEK293T cells transfected with pcDNA3 FLAG-LINCR were lysed in the lysis buffer. After centrifugation, their supernatants were immunoprecipitated with anti-FLAG antibodies (anti-FLAG affinity M2 gel; Sigma) and eluted with 250 μg/mL 3×FLAG-peptide (Sigma) in an elution buffer containing 150 mM NaCl and 20 mM Tris-HCl (pH 7.5). Eluted proteins were dialyzed against the elution buffer and subsequently preserved at −80 °C. For purification of GST-MKP1, the E. coli BL21 (DE3) (NEB) strain was introduced with pGEX6P-1 MKP1 and treated with 1 mM isopropyl-1-thio β-d-galactoside (IPTG) for 6 h at 37 °C. The recombinant proteins were extracted with a lysis buffer (20 mM Tris-HCl (pH 7.5), 0.5% Triton X-100, 150 mM NaCl, and 2 mM EDTA) and were affinity purified using glutathione Sepharose 4B beads (GE healthcare, Chicago, IL, USA). 

### 2.10. In Vitro Ubiquitination Assay

In vitro ubiquitination assays were conducted using the E2-Ubiquitin Conjugation Kit (Abcam, Cambridge, UK), according to the manufacturer’s protocol with minor modifications. Briefly, ubiquitination reactions were conducted with recombinant FLAG-LINCR as an E3 enzyme, with GST-MKP1 as a substrate, in a reaction buffer that contained recombinant biotinylated ubiquitin, His-tagged E2-conjugating enzyme, E1-activating enzyme, and Mg-adenosine triphosphate. The reaction mixtures were incubated for 8 h at 37 °C, and the reactions were stopped by the addition of 2× sample buffer; the mixtures were then subjected to immunoblot analysis with antibodies against K48-linked polyubiquitin and MKP1.

### 2.11. In Vivo Ubiquitination Assay

HEK293-TLR4 cells that expressed the indicated plasmids were treated with 5 μM MG132 for 4 h before collection, lysed in the lysis buffer containing 10 mM N-ethylmaleimide (NEM), and subjected to immunoprecipitation with anti-FLAG antibodies. The immunoprecipitates were washed with lysis buffer and then heated at 98 °C with lysis buffer containing 1% SDS to disrupt noncovalent protein–protein interactions. The heat-treated samples were evaluated by immunoblot analysis.

### 2.12. Quantitative Real-Time PCR

RAW264.7 cells seeded on 12-well plates at a density of 2 × 10^5^/well were suspended in Sepasol-RNA I Super G (Nacalai) for 15 min, and Chloroform: isoamyl alcohol (24:1, *v*/*v*) was added after indicated stimulation or treatment. Thereafter, vortexed samples were centrifuged at 15,000 rpm for 20 min at 4 °C. In this process, RNA is fractionated into a hydrophilic fraction, and DNA is fractionated into a hydrophobic fraction. RNA was precipitated by adding 2-propanol to the hydrophilic fraction; this solution was allowed to stand for 10 min and was then centrifuged at 15,000 rpm for 10 min at 4 °C. The RNA pellets were washed twice with 70% ethanol, air-dried, and then dissolved in RNase-free water. RNA purity was checked by Nanodrop ND-1000 (Thermo Fisher Scientific, Waltham, MA, USA). Total RNA was reverse transcribed using a High-Capacity cDNA Reverse Transcription Kit (Applied Biosystems, Waltham, MA, USA) according to the manufacturer’s instructions. Template cDNA was amplified by quantitative real-time PCR as described previously [30]. The primers used for the quantitative real-time PCR were 5′-GAGATCCTGTCCTTCCTGTACC-3′ and 5′-CAGCATCCTTGATGGAGTCTAT-3′ for mouse MKP1; 5′-GAAATGCCACCTTTTGACAGTG-3′ and 5′-CTGGATGCTCTCATCAGGACA-3′ for mouse IL-1β; 5′-GAGGATACCACTCCCAACAGACC-3′ and 5′-AAGTGCATCATCGTTGTTCATACA-3′ for mouse IL-6; 5′-ACTTCGGGGTGATCGGTCCCC-3′ and 5′-TGGTTTGCTACGACGTGGGCTAC-3′ for mouse TNF-α; and 5′-TGTGTCCGTCGTGGATCTGA-3′ and 5′-CCTGCTTCACCACCTTCTTGAT-3′ for mouse GAPDH. The gene expression levels were normalized to that of GAPDH.

### 2.13. Luciferase Assay

RAW264.7 cells transfected with 5× κB Firefly luciferase and thymidine kinase Renilla luciferase (Promega, Madison, WI, USA) were subjected to a luciferase assay using a Dual Luciferase Reporter Assay System (Promega) according to the manufacturer’s instructions. Firefly luciferase activity was normalized to Renilla luciferase activity.

### 2.14. Enzyme-Linked Immunosorbent Assay (ELISA)

Concentrations of mouse IL-6 and mouse TNF-α in supernatants of cell culture were measured by specific ELISA kits (Invitrogen, Madison, WI, USA) according to the manufacturer’s instructions [32].

### 2.15. Statistical Analysis

All experiments were repeated at least three independent times. The value was expressed as the mean ± standard error of the mean (S.E.M.) using Prism 9 Version 9.5.1 software (GraphPad, La Jolla, CA, USA). Two groups were compared by using Student’s *t*-test. Multiple group comparisons were performed by using the one-way ANOVA analysis of variance followed by the Tukey–Kramer test with Prism software (GraphPad). Data were considered significant when * *p* < 0.05, ** *p* < 0.01, or *** *p* < 0.001.

## 3. Results

### 3.1. LINCR Is Required for TLR4-Mediated Activation of MAP Kinase Pathways

To investigate the function of LINCR in TLR4 signaling, we established LINCR knockout (KO) cells in murine macrophage-like RAW264.7 cells by using the CRISPR/Cas9 system as previously described (Figure 1A,B) [33]. Interestingly, we found that TLR4-mediated activation of MAP kinases, such as p38 MAPK, JNK, and ERK was clearly attenuated in LINCR KO RAW264.7 cells (Figure 1C). However, the nuclear translocation of p65/RelA, an index of NF-κB activation, was not affected by the knockout of LINCR (Figure 1D). This observation was confirmed by the luciferase reporter assays of NF-κB (Figure 1E). Collectively, these observations suggest that LINCR is required for the TLR4-mediated activation of MAP kinase signaling but not for NF-κB pathways. We next investigated whether the E3 ubiquitin ligase activity of LINCR is required for the TLR4-mediated activation of MAP kinase pathways. To this end, we established LINCR-reconstituted RAW264.7 cells (Figure 1F). The reconstitution of LINCR wild type (WT) in LINCR KO RAW264.7 cells successfully restored the activation of MAP kinase pathways, whereas that of an enzymatically inactive mutant of LINCR (ΔRING domain) lacking the RING domain essential for its enzymatic activity failed to do so (Figure 1G,H). Therefore, LINCR appears to activate MAP kinase pathways through ubiquitination, which raises the possibility that LINCR leads to proteasomal degradation of inhibitory proteins of MAP kinase pathways.

### 3.2. LINCR Stimulates MAP Kinase Activation by Targeting MKP1 but Not MKP5

It has been reported that MKPs act as inhibitory proteins of MAP kinase activation by dephosphorylating MAP kinases. In particular, both MKP1 and MKP5 can dephosphorylate p38 MAPK, JNK, and ERK downstream of TLR4 [34]. To confirm the function of MKP1 and MKP5, we established MKP1 and MKP5 KO RAW264.7 cells (Figure 2A,B). As previously reported, the activation of MAP kinases was apparently enhanced in both MKP1 and MKP5 KO RAW264.7 cells (Figure 2C,D). Therefore, both MKP1 and MKP5 appear to inhibit TLR4-mediated MAP kinase activation in RAW264.7 cells. We therefore speculated that LINCR stimulates MAP kinase activation by targeting MKP1 or MKP5, and we then established LINCR/MKP1 and LINCR/MKP5 double knockout (DKO) RAW264.7 cells in order to test this possibility (Figure 2E,F). As shown in Figure 2G,H, LINCR/MKP1 DKO RAW264.7 cells exhibited the enhanced activation of MAP kinases even in the absence of LINCR, whereas LINCR/MKP5 DKO RAW264.7 cells did not. These observations mean that LINCR-mediated MAP kinase activation is overridden by the MKP1 knockout, suggesting that LINCR stimulates MAP kinase activation by targeting MKP1 but not MKP5.

### 3.3. LINCR Is Involved in the Destabilization of MKP1

We next investigated a functional link between LINCR and MKP1. As previous studies have demonstrated, TLR4 activation upregulated MKP1 at both mRNA and protein levels (Figure 3A,B). In addition, the upregulation of MKP1 mRNA and protein was clearly inhibited by the inhibitors of the ERK pathway, including the MEK (an upstream kinase of ERK) inhibitor U0126, and by the tumor progression locus 2 (TPL2) (an upstream kinase of MEK) inhibitor (Figure 3C,D) [35]. The MEK inhibitor U0126 led to the prolonged activation of p38 MAPK and JNK (Figure 3E). Therefore, the upregulation of MKP1 is mediated by the TPL2–MEK–ERK axis, which appears to prevent the prolonged or excessive activation of p38 MAPK and JNK. On the other hand, the upregulation of MKP1 mRNA was significantly reduced in LINCR KO RAW264.7 cells, which was recovered by the reconstitution of LINCR WT (Figure 3F). This observation seems to be reasonable because TLR4-mediated ERK activation was apparently reduced in LINCR KO RAW264.7 cells, as shown in Figure 1C. However, interestingly, the upregulation of MKP1 protein was enhanced in LINCR KO RAW264.7 cells, which was cancelled by the reconstitution of LINCR WT (Figure 3G,H). Moreover, the expression levels of MKP1 in LINCR KO RAW264.7 cells were higher than that of MKP1 in LINCR WT RAW264.7 cells at 15–30 min after LPS stimulation, suggesting that a small amount of LINCR under unstimulated conditions downregulates the MKP1 expression levels (Figure 3I,J). In order to explain this discrepancy that the upregulation of MKP1 protein was enhanced even though mRNA levels were downregulated in LINCR KO RAW264.7 cells, we next focused on protein expression of MKP1 regulated by LINCR. At first, we found that the expression levels of MKP1 were recovered by the proteasome inhibitor MG132, suggesting that MKP1 is continuously degraded by proteasome (Figure 3K). On the other hand, in the presence of the transcriptional inhibitor actinomycin D (ActD), MKP1 was degraded in a time course-dependent manner (Figure 3L). Of note, we found that the degradation of MKP1 is delayed in LINCR KO RAW264.7 cells, suggesting that LINCR is required for the degradation of MKP1 (Figure 3J). Collectively, these observations raise the possibility that LINCR directly degrades MKP1 through its ubiquitination.

### 3.4. LINCR Ubiquitinates MKP1

We therefore examined whether LINCR promotes the MKP1 ubiquitination. Immunoprecipitation analyses revealed the interaction between LINCR and MKP1 (Figure 4A–C). We next investigated whether LINCR directly ubiquitinates MKP1. As shown in Figure 4D, the ubiquitination of MKP1 was enhanced in the presence of LINCR. Moreover, the LINCR-dependent ubiquitination of MKP1 was observed when an in vitro ubiquitination assay was performed (Figure 4E). In order to confirm the requirement of the E3 ligase activity of LINCR, we constructed a mutant form of LINCR that was predicted to have no enzymatic activity because of mutations in the two cysteines (C202/205S) required to retain the structure of the RING domain [36]. Indeed, the mutant form of LINCR failed to induce self-ubiquitination, typically induced when LINCR is exogenously expressed [23], suggesting that the LINCR 2CS mutant has no enzymatic activity (Figure 4F). Moreover, it was shown that the LINCR 2CS mutant cannot ubiquitinate MKP1 either in cells or in vitro (Figure 4G,H). Collectively, these observations suggest that LINCR binds and ubiquitinates MKP1, leading to the proteasomal degradation of MKP1.

### 3.5. LINCR Promotes TLR-Induced Production of a Series of Inflammatory Cytokines

Given that LINCR positively regulates the activation of MAP kinase pathways mediated by TLR4 through the proteasomal degradation of MKP1, LINCR may be involved in TLR4-driven pro-inflammatory responses because MAP kinase pathways contribute to the induction of pro-inflammatory cytokines [10]. Indeed, the induction of typical pro-inflammatory cytokines, such as interleukin 1β (IL-1β), IL-6, and tumor necrosis factor-α (TNF-α), was significantly attenuated in LINCR KO RAW264.7 cells, which was recovered by the reconstitution of LINCR WT (Figure 5A–C). Consistent with these observations, ELISA assays demonstrate the requirement of LINCR to produce IL-6 and TNF-α (Figure 5D,E). Therefore, these results suggest that LINCR promotes TLR4-mediated inflammatory responses by amplifying the activation of MAP kinase pathways. Finally, we examined the functions of LINCR in other TLR signaling pathways. Interestingly, the activation of MAP kinase pathways induced by the TLR3 agonist polyinosinic-polycytidylic acid (poly(I:C)) or by the TLR5 agonist flagellin was attenuated in LINCR KO cells (Figure 5F,G). Similar to LPS stimulation (Figure 3G), the stabilization of MKP1 caused by poly(I:C) or flagellin was enhanced in LINCR KO cells (Figure 5F,G). Additionally, the induction of pro-inflammatory cytokines induced by the activation of TLR3 and TLR5 was attenuated in LINCR KO cells, which was recovered by the reconstitution of LINCR WT (Figure 5H–M). Moreover, the reduced induction of pro-inflammatory cytokines in LINCR KO cells was cancelled in LINCR/MKP1 DKO RAW264.7 cells (Figure 6A–I). We therefore conclude that LINCR amplifies inflammatory responses mediated by various TLRs through the negative regulation of MKP1 (Figure 7).

## 4. Discussion

To date, various negative regulators of TLR signals, including the deubiquitinating enzyme A20 and the phosphatase MKPs, have been identified [37,38]. However, there have been few reports on positive regulators that enhance TLR signals. In this study, we identified LINCR as a positive regulator of TLR signals that enhances the activation of MAP kinase signaling pathways through the ubiquitinating-dependent degradation of MKP1 (Figure 7). The regulation of TLR signaling by LINCR seems to be a fine-tuning mechanism of the innate immune response through crosstalk between phosphorylation and ubiquitination. Suppression of the inflammatory responses is known to increase susceptibility to infection, while excessive activation of the inflammatory responses initiates a cytokine storm, leading to systemic inflammation and multiorgan failure [39,40]. It is therefore important to induce an inflammatory response of appropriate intensity in a context-dependent manner. MKP1 is induced downstream of TLR signals and is known to function as a major negative regulator of inflammatory responses, whereas LINCR cancels this suppression mediated by MKP1 and thereby seems to induce sufficient activation of inflammatory responses. However, as shown in Figure 3L, the degradation of MKP1 was not completely suppressed in LINCR KO cells. This observation suggests that MKP1 is degraded by LINCR-independent mechanisms and that the expression level of MKP1 is coordinately controlled not only by LINCR but also by other E3 ligases, including Skp2 and atrogin-1 [20,21], despite LINCR seemingly acting as a main E3 ligase for MKP1 degradation in this context. On the other hand, recent evidence has demonstrated that the formation of K48/K63-mixed chains enhance the proteasomal degradation of substrates [41]. We therefore investigated the involvement of K48/K63-mixed chains in LINCR-mediated ubiquitination of MKP1, but we were unable to detect K63-linked chains on MKP1. Rather, a previous report raises the possibility that LINCR adds K11-linked ubiquitin to MKP1 [23]. In any case, this study is one of the few reports on positive regulators of TLR4 signaling; however, a limitation of this study is that the function of LINCR was evaluated only in RAW264.7 cells. Therefore, further studies using in vivo experiments are required to show the physiological role of LINCR.

Since the expression levels of MKP1 are downregulated in some inflammatory diseases such as colitis and psoriasis, upregulation of MKP1 activation or expression has been suggested to be beneficial as a treatment for inflammatory diseases [42,43]. Glucocorticoids (GCs) bind to glucocorticoid receptors (GRs) and exert anti-inflammatory effects through promoting the biosynthesis of anti-inflammatory factors. Indeed, MKP1 is included in the targets of GCs, and dexamethasone—a type of GC—does in fact increase the expression of MKP1 [44]. Moreover, the therapeutic effect of dexamethasone against endotoxin shock is reduced in MKP1 KO mice, suggesting that increased expression of MKP1 is required for the anti-inflammatory effects of dexamethasone [44,45]. Interestingly, LINCR was discovered as a protein whose LPS-dependent induction is suppressed by GCs [23]. Our results revealed that decreased expression of LINCR leads to an increase in MKP1, suggesting that the silence of LINCR contributes to the anti-inflammatory effect of GCs. On the other hand, GCs frequently initiate side effects, such as susceptibility to infection and adrenal insufficiency, because GCs non-specifically inhibit immune responses. Our results demonstrate that the silence of LINCR more specifically causes MKP1 accumulation. Moreover, our findings that LINCR is required for the immune responses induced by the activation of various TLRs (Figure 5F–M) raise the possibility that LINCR may be useful as a new therapeutic target for inflammatory diseases.

## Figures and Tables

**Figure 1 cells-13-00687-f001:**
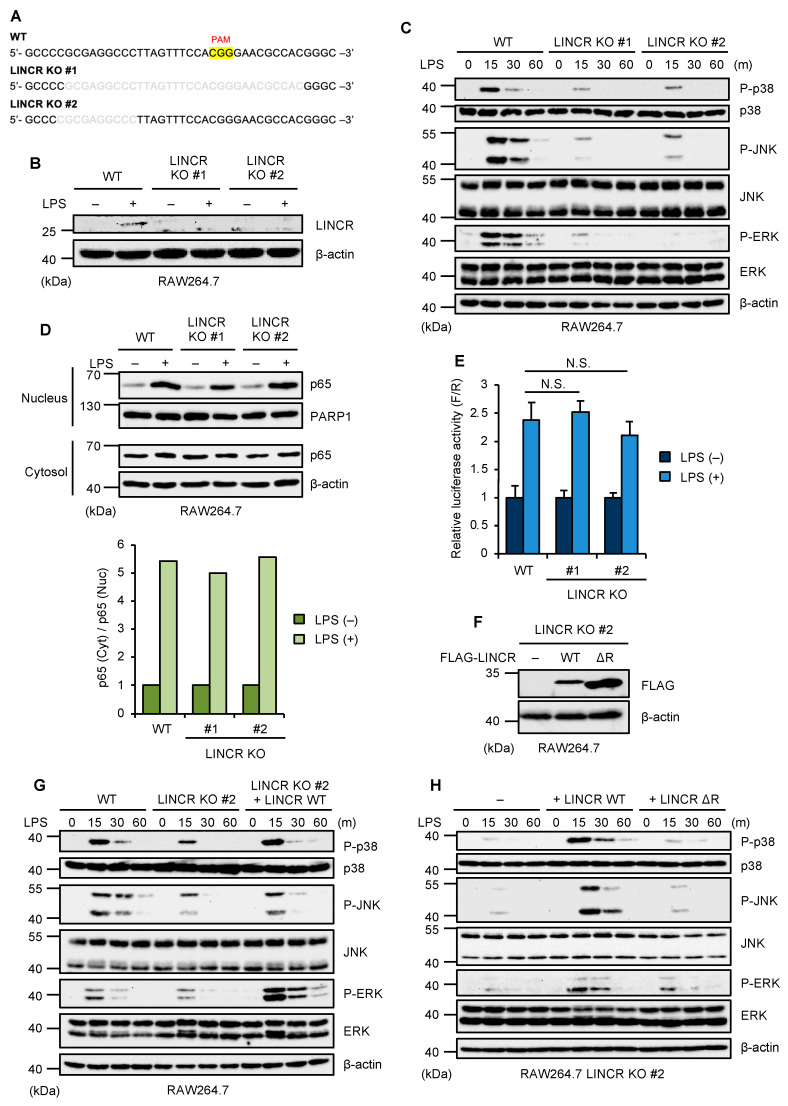
LINCR promotes LPS-induced activation of MAP kinases. (**A**) DNA sequences around the guide RNA (gRNA) target site in the exon 2 of the LINCR WT and KO RAW264.7 cells. (**B**) WT and LINCR KO RAW264.7 cells were treated with LPS (100 ng/mL) for 2 h. The cell lysates were subjected to immunoblotting with the indicated antibodies. (**C**) WT and LINCR KO RAW264.7 cells were treated with LPS (100 ng/mL) for the indicated time. The cell lysates were subjected to immunoblotting with the indicated antibodies. (**D**) WT and LINCR KO RAW264.7 cells were treated with LPS (100 ng/mL) for 2 h. The cell lysates were subjected to immunoblotting with the indicated antibodies. Relative amounts of nuclear p65 were calculated after normalizing cytoplasmic and nuclear p65 with PARP1 or β-actin, respectively, and are shown in the lower panel. (**E**) WT and LINCR KO RAW264.7 cells were transfected with a plasmid and a Renilla luciferase plasmid for normalization. After 24 h, cells were treated with LPS (100 ng/mL) for 2 h. Firefly and Renilla luciferase activities were quantified with a dual luciferase assay kit. Graphs are shown as mean ± S.D. (*n* = 3). Statistical significance was determined by one-way ANOVA followed by Tukey’s test; N.S.: not significant. (**F**) Immunoblot analysis in LINCR KO, LINCR-reconstituted, and LINCR ΔRING-reconstituted RAW264.7 cells. The cell lysates were subjected to immunoblotting with the indicated antibodies. (**G**) WT, LINCR KO, and LINCR-reconstituted RAW264.7 cells were treated with LPS (100 ng/mL) for the indicated time. The cell lysates were subjected to immunoblotting with the indicated antibodies. (**H**) LINCR KO, LINCR-reconstituted, and LINCR ΔRING-reconstituted RAW264.7 cells were treated with LPS (100 ng/mL) for the indicated time. The cell lysates were subjected to immunoblotting with the indicated antibodies.

**Figure 2 cells-13-00687-f002:**
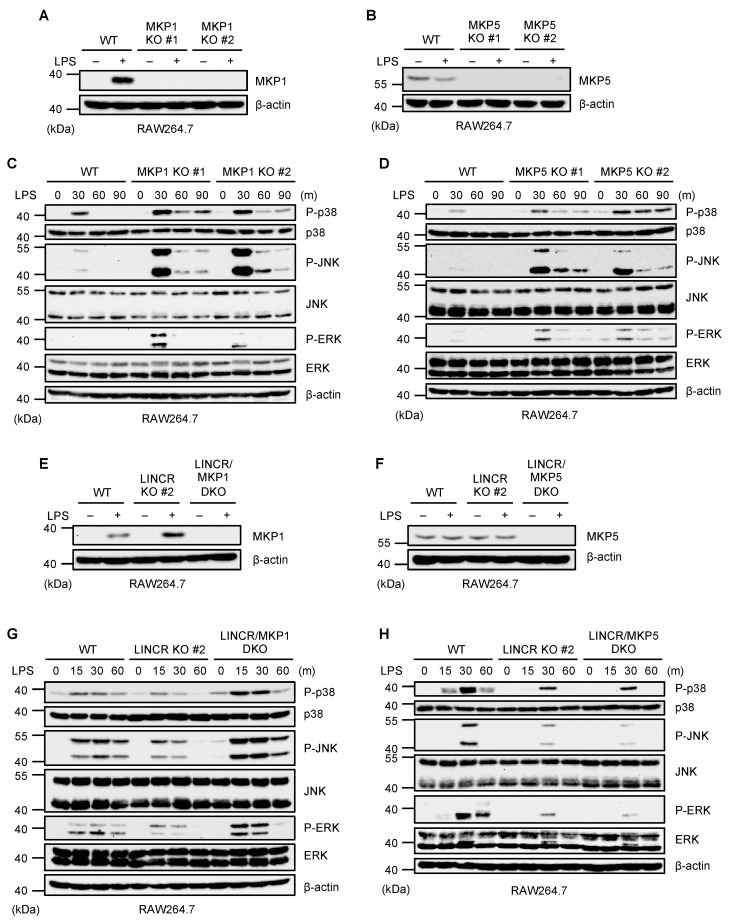
LINCR promotes LPS-induced activation of MAP kinases in an MKP1-dependent manner. (**A**,**B**) WT and MKP1 KO (**A**) or MKP5 KO (**B**) RAW264.7 cells were treated with LPS (100 ng/mL) for 2 h. The cell lysates were subjected to immunoblotting with the indicated antibodies. (**C**,**D**) WT and MKP1 KO (**C**) or MKP5 KO (**D**) RAW264.7 cells were treated with LPS (100 ng/mL) for the indicated time. The cell lysates were subjected to immunoblotting with the indicated antibodies. (**E**,**F**) LINCR KO and LINCR/MKP1 DKO (**E**) or LINCR/MKP5 DKO (**F**) RAW264.7 cells were treated with LPS (100 ng/mL) for 2 h. The cell lysates were subjected to immunoblotting with the indicated antibodies. (**G**,**H**) WT and LINCR KO along with LINCR/MKP1 DKO (**G**) or along with LINCR/MKP5 DKO (**H**) RAW264.7 cells were treated with LPS (100 ng/mL) for the indicated time. The cell lysates were subjected to immunoblotting with the indicated antibodies.

**Figure 3 cells-13-00687-f003:**
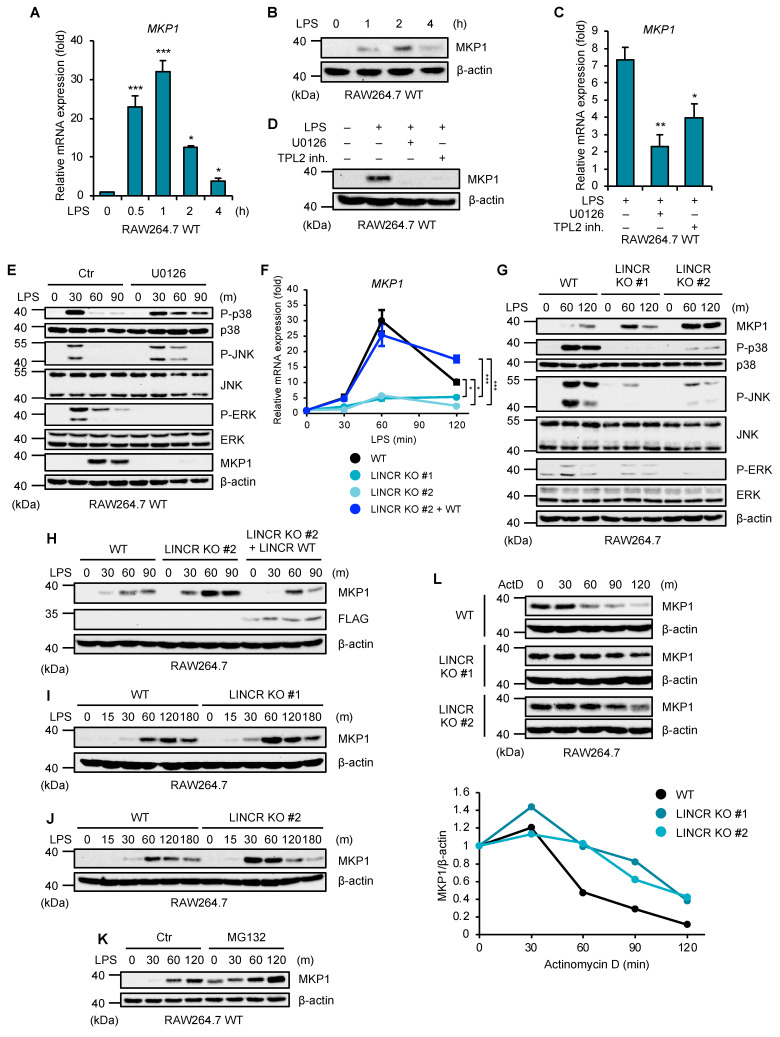
LINCR inhibits LPS-induced accumulation of MKP1. (**A**) RAW264.7 cells were treated with LPS (100 ng/mL) for the indicated time. The mRNA levels of MKP1 were analyzed by quantitative real-time PCR (normalized with GAPDH mRNA levels). Graphs are shown as mean ± S.D. (*n* = 3). Statistical significance was determined by one-way ANOVA followed by Tukey’s test; * *p* < 0.05, *** *p* < 0.001. (**B**) RAW264.7 cells were treated with LPS (100 ng/mL) for the indicated time. The cell lysates were subjected to immunoblotting with the indicated antibodies. (**C**) RAW264.7 cells were treated with DMSO, U0126 (10 μM), or TPL2 inhibitor (10 μM) for 30 min and then with LPS (100 ng/mL) for 2 h. The mRNA levels of MKP1 were analyzed by quantitative real-time PCR (normalized with GAPDH mRNA levels). Graphs are shown as mean ± S.D. (*n* = 3). Statistical significance was determined by one-way ANOVA followed by Tukey’s test; * *p* < 0.05, ** *p* < 0.01. (**D**) RAW264.7 cells were treated with DMSO, U0126 (10 μM), or TPL2 inhibitor (20 μM) for 30 min and then with LPS (100 ng/mL) for 2 h. The cell lysates were subjected to immunoblotting with the indicated antibodies. (**E**) RAW264.7 cells were treated with DMSO or U0126 (10 μM) for 30 min and then with LPS (100 ng/mL) for the indicated time. The cell lysates were subjected to immunoblotting with the indicated antibodies. (**F**) WT, LINCR KO, and LINCR-reconstituted RAW264.7 cells were treated with LPS (100 ng/mL) for the indicated time. The mRNA levels of MKP1 were analyzed by quantitative real-time PCR (normalized with GAPDH mRNA levels). Graphs are shown as mean ± S.D. (*n* = 3). Statistical significance was determined by one-way ANOVA followed by Tukey’s test; * *p* < 0.05, *** *p* < 0.001. (**G**,**I**,**J**) WT and LINCR KO RAW264.7 cells were treated with LPS (100 ng/mL) for the indicated time. The cell lysates were subjected to immunoblotting with the indicated antibodies. (**H**) WT, LINCR KO, and LINCR-reconstituted RAW264.7 cells were treated with LPS (100 ng/mL) for the indicated time. The cell lysates were subjected to immunoblotting with the indicated antibodies. (**K**) RAW264.7 cells were treated with DMSO or MG132 (5 μM) for 30 min and then with LPS (100 ng/mL) for the indicated time. The cell lysates were subjected to immunoblotting with the indicated antibodies. (**L**) RAW264.7 cells were treated with LPS (100 ng/mL) for 30 min and then with actinomycin D (1 μg/mL) for the indicated time. The cell lysates were subjected to immunoblotting with the indicated antibodies. Relative amounts of MKP1 were calculated after normalizing to β-actin and are shown in the lower panel.

**Figure 4 cells-13-00687-f004:**
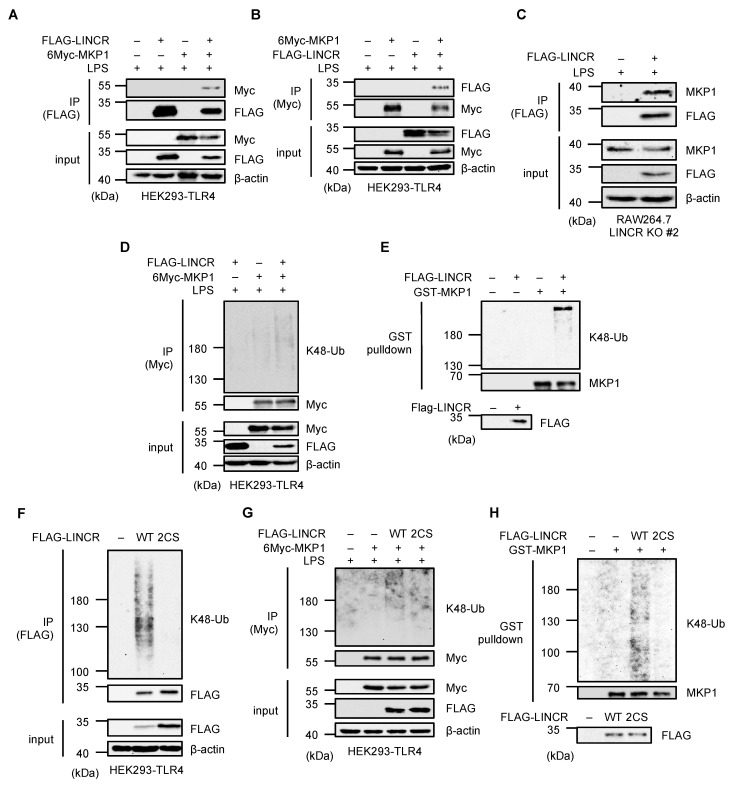
LINCR promotes K48-linked polyubiquitination of MKP1. (**A**) HEK293-TLR4 cells transfected with the indicated constructs were treated with LPS (100 ng/mL) for 2 h. The cell lysates were immunoprecipitated with an anti-FLAG antibody and subjected to immunoblotting with the indicated antibodies. (**B**) HEK293-TLR4 cells transfected with the indicated constructs were treated with LPS (100 ng/mL) for 2 h. The cell lysates were immunoprecipitated with an anti-Myc antibody and subjected to immunoblotting with the indicated antibodies. (**C**) LINCR KO and FLAG-LINCR-reconstituted RAW264.7 cells were treated with LPS (100 ng/mL) and MG132 (5 μM) for 2 h. The cell lysates were immunoprecipitated with an anti-FLAG antibody and subjected to immunoblotting with the indicated antibodies. (**D**,**G**) HEK293-TLR4 cells were transfected with the indicated constructs and subjected to an in vivo ubiquitination assay to assess 6Myc-MKP1 ubiquitination status. (**E**,**H**) An in vitro ubiquitination assay using purified recombinant proteins. Ubiquitination reactions were performed using the recombinant proteins and subjected to immunoblotting with the indicated antibodies. (**F**) HEK293-TLR4 cells were transfected with the indicated constructs and subjected to an in vivo ubiquitination assay to assess FLAG-LINCR ubiquitination status.

**Figure 5 cells-13-00687-f005:**
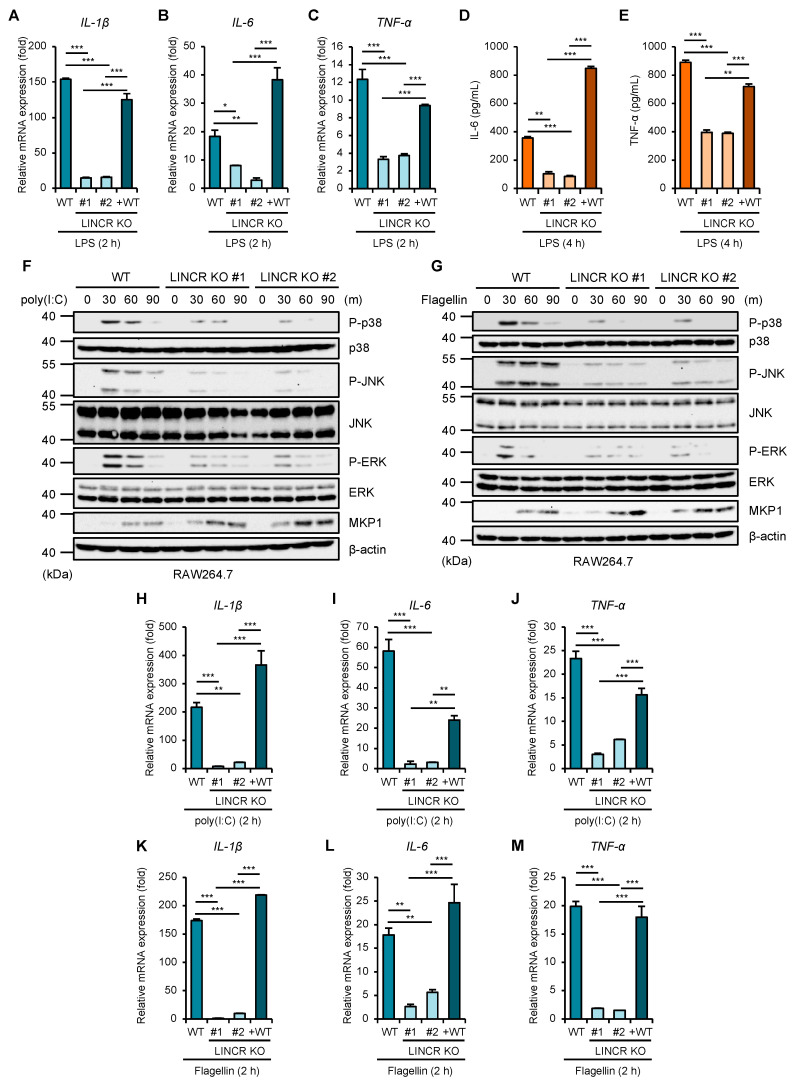
LINCR promotes LPS-induced production of inflammatory cytokines. (**A**–**C**) WT, LINCR KO, and LINCR-reconstituted RAW264.7 cells were treated with LPS (100 ng/mL) for 2 h. The mRNA levels of IL-1β (**A**), IL-6 (**B**), and TNF-α (**C**) were analyzed by quantitative real-time PCR (normalized with GAPDH mRNA levels). Graphs are shown as mean ± S.D. (*n* = 3). Statistical significance was determined by Student’s *t*-test; * *p* < 0.05, ** *p* < 0.01, *** *p* < 0.001. (**D**,**E**) ELISA analysis of cytokines in culture supernatants of RAW264.7 cells. RAW264.7 cells were treated with LPS (100 ng/mL) for the 2 h. IL-6 (**D**) or TNF-α (**E**) releases were analyzed by ELISA. Graphs are shown as mean ± S.D. (*n* = 3). Statistical significance was determined by Student’s *t*-test; ** *p* < 0.01, *** *p* < 0.001. (**F**,**G**) WT and LINCR KO RAW264.7 cells were treated with poly(I:C) (1 μg/mL) (**F**) or Flagellin (1 μg/mL) (**G**) for the indicated time. The cell lysates were subjected to immunoblotting with the indicated antibodies. (**H**–**M**) WT, LINCR KO, and LINCR-reconstituted RAW264.7 cells were treated with poly(I:C) (1 μg/mL) (**H**–**J**) or Flagellin (1 μg/mL) (**K**–**M**) for 2 h. The mRNA levels of IL-1β (**H**,**K**), IL-6 (**I**,**L**) and TNF-α (**J**,**M**) were analyzed by quantitative real-time PCR (normalized with GAPDH mRNA levels). Graphs are shown as mean ± S.D. (*n* = 3). Statistical significance was determined by Student’s *t*-test; * *p* < 0.05, ** *p* < 0.01, *** *p* < 0.001.

**Figure 6 cells-13-00687-f006:**
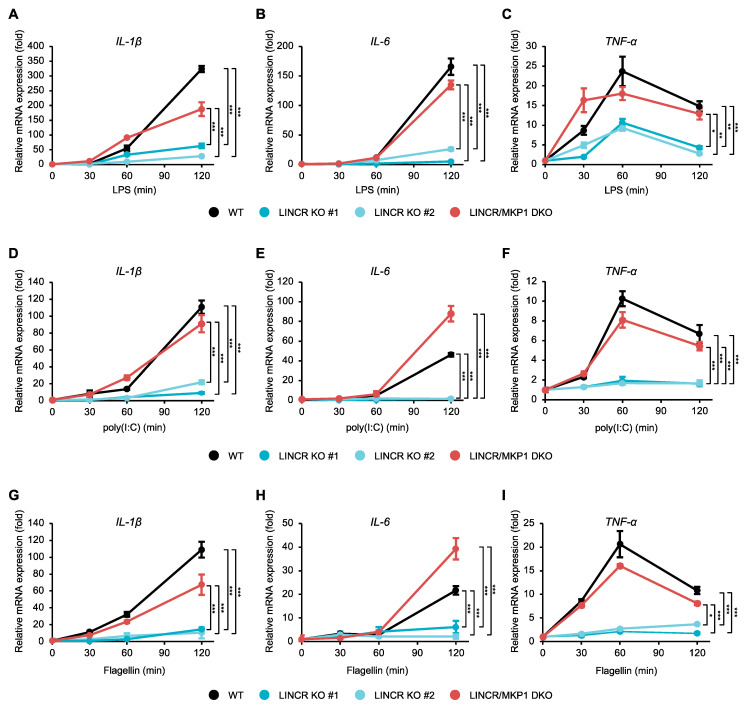
LINCR promotes TLR-induced production of inflammatory cytokines in an MKP1-dependent manner. (**A**–**I**) WT, LINCR KO, and LINCR-reconstituted RAW264.7 cells were treated with LPS (100 ng/mL) (**A**–**C**), poly(I:C) (1 μg/mL) (**D**–**F**), or Flagellin (1 μg/mL) (**G**–**I**) for the indicated time. The mRNA levels of IL-1β (**A**,**D**,**G**), IL-6 (**B**,**E**,**H**), and TNF-α (**C**,**F**,**I**) were analyzed by quantitative real-time PCR (normalized with GAPDH mRNA levels). Graphs are shown as mean ± S.D. (*n* = 3). Statistical significance was determined by one-way ANOVA followed by Tukey’s test; * *p* < 0.05, ** *p* < 0.01, *** *p* < 0.001.

**Figure 7 cells-13-00687-f007:**
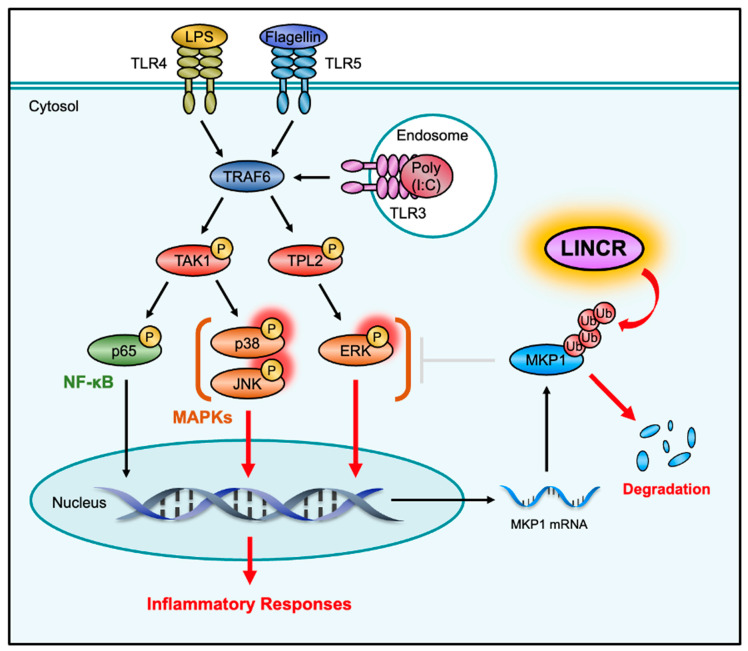
Schematic model to explain our study is described. The activation of TLRs initiates the inflammatory responses by activating MAP kinase and NF-κB signaling pathways. MKP1 negatively regulates the inflammatory responses through the dephosphorylation of MAP kinases, including JNK and p38 MAPK. In this study, we identified LINCR as a positive regulator of TLR-induced inflammatory responses and demonstrated that LINCR amplifies pro-inflammatory signals by promoting the degradation of MKP1.

## Data Availability

The data presented in this study are available in article.

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
