# Peer review of "The E3 Ubiquitin Protein Ligase LINCR Amplifies the TLR-Mediated Signals through Direct Degradation of MKP1"

_cells, 2024, doi:10.3390/cells13080687_

Round 1

Reviewer 1 Report

Comments and Suggestions for Authors

In this paper, authors identify LINCR as a novel ubiquitin ligase that regulates MKP1 phosphatase degradation and contributes to the repression of a wide range of TLRs-dependent MAPKs signaling, including TLR4. In recent years, it has become clear that the induction of MKP1 expression and its stabilization at the protein level is regulated by a variety of signals, and is associated with a variety of disease states. Thus, this is an important paper providing new insights into the high-profile research area.

Even so, this paper requires some corrections and additions for publication, which are described below.

Major comment

1. Throughout the paper, statistical analysis including Fig. 1E, multiple testing should use other methods such as ANOVA post-hoc Tukey test instead of t-test.

2. What reporter was used for the assay in Fig. 1E?. Is "3xIRS" in the Method appropriate for evaluating NF-kB activity?

3. In this paper, LINCR and MKP1 proteins are not induced (or not expressed) when unstimulated (or for a short period of time, <2 hours after LPS stimulation). However, MAPK signaling activities were changed in these KO cells. Why is this? Are LINCR and MKP1 expressed, albeit in trace amounts, even when unstimulated, and do they suppress signaling? Are there any data in the literature to support this idea? (Or, is it possible to confirm this by enriching the relevant proteins by IP or other means?)

4. The pre-treatment conditions (time), including Fig. 3D, are not specified.

5. Fig. 2E and F do not show WT cell control.

6. Is it possible to evaluate the time course including unstimulated time in Fig. 3F as in Fig. 3A? (Does LINCR have any effect on the mRNA level of short-time stimulation, including unstimulated?) Similarly, for Fig. 5A-E, H-M, does LINCR-KO affect their induction at unstimulated or short time?

7. In Fig. 4D/ E/ G, does LINCR mediate K63-Ubiquitination of MKP1, in addition to K48-Ub? Is it possible to evaluate this experimentally?

(Since literature suggests that K48/K63 (or K11)-mixed chains may further promote proteasomal degradation)

8. By what mechanisms is LINCR induced under TLR signaling, e.g. mRNA induction or stabilization at the protein level, and is there any relevant literature information?

Reviewer 2 Report

Comments and Suggestions for Authors

The authors attempted to demonstrate the functions of LINCR on the MAPK1-regulated TLR signaling by its ubiquitin ligase activity. LINCR interacted with MAPK1 and decreased MAPK1 protein levels. LINCR KO resulted in reduction of TLR-stimulated cytokine production in Raw264.7 cells. This is clearly presented study and in general the data support the major conclusions. Below the reviewer provides comments aimed at improving what is a potentially very interesting study.

1) The authors used the CRISPR/Cas9 system to generate KO cell lines. It seems that they have not used single clones for this study. If they used bulk cells, they should show the efficacy of indel mutations in each KO cell line. Also, they used WT as control for all experiments. What is “WT” implying? If it is parental Raw264.7 cells, they should use non-targeting sgRNA as negative controls. 

2) In figure 1D, nuclear p65 vs cytoplasmic p65 ratio looks difference among the three conditions. If authors say no difference among the samples, show a nuclear/cytoplasmic intensity ratio. Is 3xIRS firefly Luc specific for NF-kB? If so, refer proper papers. I couldn’t reach the original paper to check it.

3) Why are the peaks of phosphorylation of p38, JNK and ERK delayed in Fig. 1H compared to the other experiments?

4) The peak of MKP1 mRNA expression is 0.5 to 1 h after LPS stimulation. The authors compared the MKP1 expression at 2 h after LPS treatment (Fig 3F). Are there any differences at early timepoint (0.5 or 1 h)?

5) The authors mentioned MPK1 is continuously degraded, and the data support it. However, the expression of LINKCR is not detectable in the absence of LPS (Fig. 1B). Also, LINCR KO resulted in a delay in the decrease of MKP1 protein rather than a complete inhibition of degradation. These suggest that MPK1 is degraded not only by LINCR but the other mechanisms. Discuss about it.

6) The authors demonstrated LINCR induced self-ubiquitination (Fig. 4F). In Figures 4, they detected the polyubiquitination using anti K48-Ub antibody. Are these ubiquitinated proteins MKP1? MKP1 makes a complex with LINCR, which mean MKP1 immunoprecipitation contains LINCR protein. Immunoprecipitation with anti-Ub antibody and detecting with MKP1 may help to answer this question. Or use a MKP1 mutant which lacks Ub sites. 

7) LINCR KO decreased cytokine productions after TLR ligands stimulation (Figures 5), and the authors state that “LINCR promotes the activation of the MAP kinase pathways through the K48-linked polyubiquitination and subsequent proteasomal degradation of MKP1, leading to the enhanced production of pro-inflammatory cytokines.”. LINCR KO definitely changed the cytokine production, but the effect of MKP1 on cytokine production was not examined. If they want to say LINCR/MKP1 regulates cytokine production, need to use DKO cells. 

8) In this study, they used Raw264.7 cells for most of experiments. Therefore, the regulation of the TLR-mediated signals by LINCR and MKP1 is limited to Raw264.7 cells. It is better to emphasize it. 

Round 2

Reviewer 1 Report

Comments and Suggestions for Authors

The authors have adequately responded to all comments, and the manuscript appears to be at the level required for publication in this journal.

Author Response

We would like to thank Reviewer #1 for useful advice on how to improve the manuscript.

Reviewer 2 Report

Comments and Suggestions for Authors

The authors have addressed most of my concerns and the manuscript is greatly improved. However, I still have a few points I am not convinced of.

1. If they want to consider genetic background, they should use same intact Raw264.7 cells for making non-targeting sgRNA transduced cells. My concern is the effect of sgRNA transduction and lentiviral infection on cytokine production. It is not necessary to redo all experiments, they can compare WT and non-targeting sgRNA transduced Raw cells. Alternatively, they can refer to previous reports using WT and sgRNA-transduced Raw cells.

2. Regarding the detection of polyubiquitinating proteins, the immunoprecipitated or GST pulled-down samples should have both MKP1 and LINCR. They have clearly showed the interaction between MKP1 and LINCR using co-immunoprecipitation assay. Boiling with 1 % SDS can break the protein-protein interaction, but the proteins remain in the same sample buffer. Therefore, I think their response is not convincing. 

Round 3

Reviewer 2 Report

Comments and Suggestions for Authors

The revised version progressed critiques well, and this reviewer does not have any more questions and critiques.

Author Response

We would like to thank Reviewer #2 for useful advice again.